# An Archimedes' screw for light

Emanuele Galiffi [1✉], Paloma A. Huidobro [2] & J. B. Pendry [3✉]

An Archimedes' Screw captures water, feeding energy into it by lifting it to a higher level. We introduce the first instance of an optical Archimedes' Screw, and demonstrate how this system is capable of capturing light, dragging it and amplifying it. We unveil new exact analytic solutions to Maxwell's Equations for a wide family of chiral space-time media, and show their potential to achieve chirally selective amplification within widely tunable parity-time-broken phases. Our work, which may be readily implemented via pump-probe experiments with circularly polarized beams, opens a new direction in the physics of time-varying media by merging the rising field of space-time metamaterials and that of chiral systems, and offers a new playground for topological and non-Hermitian photonics, with potential applications to chiral spectroscopy and sensing.

[1] Advanced Science Research Center, City University of New York, 85 St. Nicholas Terrace, New York, NY 10031, USA. [2] Instituto de Telecomunicações, Instituto Superior Tecnico-University of Lisbon, Avenida Rovisco Pais 1, 1049-001 Lisboa, Portugal. [3] The Blackett Laboratory, Department of Physics, Imperial College London, South Kensington Campus, London SW7 2AZ, UK. ✉email: eg612@ic.ac.uk; j.pendry@imperial.ac.uk

Fundamental aspects of wave interactions in time-dependent systems have recently attracted renewed interest, thanks to the discovery of ultrathin and highly nonlinear materials. Freed from constraints such as reciprocity and energy conservation, these systems can enable new and exotic wave behaviours. In this work we open a new direction in the rising field of space-time metamaterials by blending it for the first time with the established field of chiral systems, realising the electromagnetic analogue of the famous Archimedes' screw for fluids.

The significance of time-varying media for wave manipulation rose from the several proposals amidst the decade-long quest for achieving magnet-free nonreciprocity both in photonics[1–3] and with mechanical waves[4,5]. Temporal structuring of matter opens several new avenues for wave control: periodic modulations of material parameters can enable the design of topologically nontrivial phases[6] as well as Floquet topological insulators[7] and topological insulators with synthetic frequency dimensions[8]. In addition, appropriate tailoring of the temporal dependence of reactive elements can enable arbitrary energy accumulation[9], whereas the introduction of time-modulated, non-Hermitian elements can lead to nonreciprocal mode-steering and gain[10], as well as event cloaking and perfect absorption[11], and surface-wave coupling on spatially flat interfaces[12]. In non-periodic systems, abrupt switching holds the key to new directions such as time-reversal[13], time-refraction[14] and anisotropy-induced wave routing[15], as well as frequency conversion[16–18], bandwidth enhancement[19] and Anderson localization[20].

Furthermore, drawing from the combination of spatial and temporal degrees of freedom, space-time metamaterials, whose parameters are modulated in a travelling wave-type fashion[21–25], have recently acquired renewed momentum both for fundamental reasons, as they enable the mimicking and generalization of physical motion beyond the common relativistic constraints, leading to optical drag[26], localization[27] and novel amplification mechanisms[28,29], and for practical applications such as harmonic generation[30], beam steering[31] and power combination from multiple sources[32]. Successful experiments with spatiotemporal modulation include works in acoustics[5,7,33] and elasticity[34], microwaves[3,30], in the infrared[35] and even in diffusive systems[36], and they have recently started pushing closer to the optical domain[37] thanks to the introduction of novel highly nonlinear materials such as ITO[38] and AZO[39]. Finally, homogenization schemes have recently been developed for both temporal[40,41] and spatiotemporal[42] metamaterials.

A longer-established, yet still rampant, multidisciplinary field of research is that of chiral systems (we note that the term "chiral" is also used to signify a medium with bianisotropic coupling. Here, however, we only refer to its helical character, and associated circular dichroism properties). Owing to its crucial technological applications, ranging from display technology to spectroscopy and biosensing, the mathematical study of chiral electromagnetic systems dates back several decades[43], with experimental observations of optical activity dating much farther back to the early observations of Biot and Pasteur in the 19th century[44]. Theories of chiral media have been successfully applied to the study of cholesteric liquid crystals[45], as well as a variety of naturally occurring structures[46] and, since the advent of metamaterials, to negative refraction[47–49], broadband and enhanced optical activity[50], asymmetric transmission[51–53] and, more recently, topology[54].

In this work, we combine the essential ingredients of these two dominant themes of the current metamaterial scene, chirality and time-modulation, to propose the first instance of chiral space-time metamaterials, realising the electromagnetic analogue of the famous Archimedes' Screw for fluids. In developing our exact analytical model for a wide class of these systems, we uncover closed-form analytic solutions to Maxwell's Equations, and use them to demonstrate the potential of these structures for chirally selective amplification resulting from Parity-Time (PT)-broken phases. The richness of our analytic model paves the way to future systematic studies of chiral space-time media as a new playground for topological and non-Hermitian physics, and may be realized in the near future both in optics, via pump-probe experiments with circularly polarized pump beams, and at RF, with nonlinear circuit elements.

## Results and discussion

**Formalism.** Consider a medium with the following anisotropic permittivity and permeability tensors:

$$\hat{\varepsilon}/\varepsilon_1 = \mathbb{I} + \hat{\delta\varepsilon} = \mathbb{I} + 2\alpha_\varepsilon \mathbb{R}_-^T \hat{\mathbf{x}}\hat{\mathbf{x}}^T \mathbb{R}_- \tag{1}$$

$$\hat{\mu}/\mu_1 = \mathbb{I} + \hat{\delta\varepsilon} = \mathbb{I} + 2\alpha_\mu \mathbb{R}_+^T \hat{\mathbf{y}}\hat{\mathbf{y}}^T \mathbb{R}_+ \tag{2}$$

where $\hat{\mathbf{x}}$ and $\hat{\mathbf{y}}$ are unit vectors in the plane perpendicular to the propagation axis of the screw, $2\alpha$ is the modulation amplitude of the respective electromagnetic parameter, $\varepsilon_1$ and $\mu_1$ are the background permittivity and permeability of the medium, and the rotation matrix

$$\mathbb{R}_\pm = \begin{pmatrix} c(\theta^\pm) & s(\theta^\pm) & 0 \\ -s(\theta^\pm) & c(\theta^\pm) & 0 \\ 0 & 0 & 1 \end{pmatrix} \tag{3}$$

describes ($c=$ "cos" and $s=$ "sin") the screwing operation along the spatiotemporal variable $\theta^\pm = gz - \Omega t \pm \phi$. Note that we have chosen units such that $\varepsilon_0 = \mu_0 = c_0 = 1$. The wavenumber $g$ and frequency $\Omega$ of the modulation define the screw velocity $v_s = \Omega/g$, and the electric and magnetic components of the screw are separated by a dephasing $2\phi$, such that the system is impedance-matched everywhere if $\alpha_\varepsilon = \alpha_\mu$ and $\phi = 0$. Figure 1(b) depicts the tip of the principal axes (eigenvectors) $\overrightarrow{\delta\varepsilon}$ and $\overrightarrow{\delta\mu}$ of the modulated part of the material tensors for $\phi = 0$, which correspond to the respective screwing coordinates:

$$\overrightarrow{x'} = c(\theta)\overrightarrow{x} + s(\theta)\overrightarrow{y} \quad \overrightarrow{y'} = -s(\theta)\overrightarrow{x} + c(\theta)\overrightarrow{y} \tag{4}$$

The complete form of the material tensors is:

$$\hat{\varepsilon}/\varepsilon_1 = \mathbb{I} + 2\alpha_\varepsilon \begin{pmatrix} c^2(\theta^-) & c(\theta^-)s(\theta^-) & 0 \\ c(\theta^-)s(\theta^-) & s^2(\theta^-) & 0 \\ 0 & 0 & 0 \end{pmatrix} \tag{5}$$

$$\hat{\mu}/\mu_1 = \mathbb{I} + 2\alpha_\mu \begin{pmatrix} s^2(\theta^+) & -c(\theta^+)s(\theta^+) & 0 \\ -c(\theta^+)s(\theta^+) & c^2(\theta^+) & 0 \\ 0 & 0 & 0 \end{pmatrix} \tag{6}$$

For simplicity, we assume to be working in a regime where material dispersion is negligible, and focus on the normal-incidence case $k_x = k_y = 0$. In order to write an eigenvalue problem for the eigenfrequencies $\omega(k)$, we use Maxwell's Equations for the displacement field **D** and magnetic induction **B**:

$$\frac{\partial \mathbf{D}}{\partial t} = \nabla \times \hat{\mu}^{-1}\mathbf{B} \quad \frac{\partial \mathbf{B}}{\partial t} = -\nabla \times \hat{\varepsilon}^{-1}\mathbf{D}. \tag{7}$$

In order to seek an analytic solution for normal incidence, we transform our fields into a new, coordinate-dependent basis of forward ( $\rightarrow$ )- and backward ( $\leftarrow$ )-propagating fields:

$$F_{\overrightarrow{x'}} = c(\theta)(D_x + \bar{B}_y) + s(\theta)[D_y + (-\bar{B}_x)] \tag{8}$$

$$F_{\overrightarrow{y'}} = -s(\theta)(D_x + \bar{B}_y) + c(\theta)[D_y + (-\bar{B}_x)] \tag{9}$$

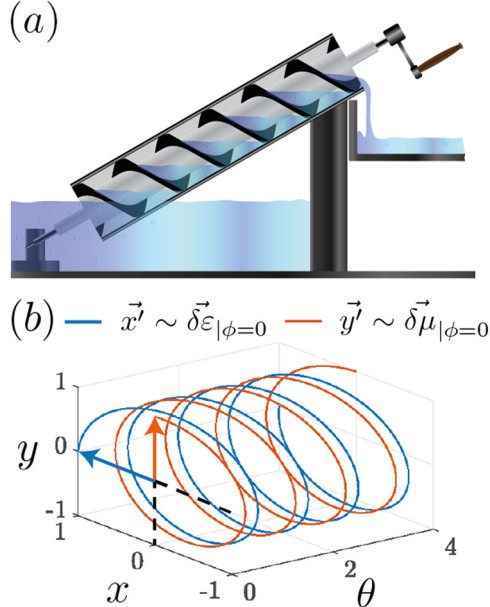

**Fig. 1 Illustration of an optical Archimedes' screw. a** A mechanical Archimedes screw carries fluids from a lower to a higher ground[56]. **b** An optical screw is a medium whose permittivity and permeability tensors are modulated such that the principal axes $\vec{\delta\varepsilon}$ and $\vec{\delta\mu}$ of their modulation describe two helices. Furthermore, in our model we allow for a dephasing $2\phi$ between the two modulations. For zero-dephasing ($\phi = 0$), at $\theta = gx - \Omega t = 0$, $\vec{\delta\varepsilon}$ points along the x-axis while $\vec{\delta\mu}$ points along the y-axis. For finite dephasing $\phi \neq 0$ the two modulations are shifted forward and backward in space-time by $2\phi$, so that their total phase difference is $2\phi$.

$$F_{x'}^{\leftarrow} = c(\theta)(D_x - \bar{B}_y) + s(\theta)[D_y - (-\bar{B}_x)] \tag{10}$$

$$F_{y'}^{\leftarrow} = -s(\theta)(D_x - \bar{B}_y) + c(\theta)[D_y - (-\bar{B}_x)] \tag{11}$$

which follow the screw symmetry $(x', y')$ of the system. Here $\bar{B}_{x/y} = \sqrt{\varepsilon_1/\mu_1}B_{x/y} = B_{x/y}/Z_1$, where $Z_1$ is the wave impedance of the background medium. Remarkably, thanks to this symmetry operation we can completely absorb into the new basis fields the $\theta$-dependence of Maxwell's Equations induced by the screw modulation, so that the infinite system of coupled equations decouples into independent 4-by-4 blocks for any dephasing $\phi$ (see SM for details). Upon assuming a Floquet-Bloch ansatz $\Psi = e^{i(kz-\omega t)}\sum_m a_m e^{i(2m-1)(gz-\Omega t)}$, where $m \in \mathbb{Z}$, each 4-by-4 block can be written as an eigenvalue problem for the $n^{th}$ set of four bands:

$$\omega_n \begin{pmatrix} \mathbf{F}_n^{\rightarrow} \\ \mathbf{F}_n^{\leftarrow} \end{pmatrix} = \begin{pmatrix} \mathbb{M}_{\rightarrow,n}^{\rightarrow} & \mathbb{M}_{\rightarrow,n}^{\leftarrow} \\ \mathbb{M}_{\leftarrow,n}^{\rightarrow} & \mathbb{M}_{\leftarrow,n}^{\leftarrow} \end{pmatrix} \begin{pmatrix} \mathbf{F}_n^{\rightarrow} \\ \mathbf{F}_n^{\leftarrow} \end{pmatrix} \tag{12}$$

where $n = 2m - 1$ is an odd integer, and the four 2-by-2 matrices $\mathbb{M}$ coupling forward and backward waves are given in closed form in the SM, together with a detailed derivation of the basis transformation. The motivation for the doubly periodic harmonic form of this Fourier expansion is that, due to the squared trigonometric functions in Eqs. (5)–(6) the actual spatiotemporal period of the system is halved. In addition, the presence of the trigonometric functions of $\theta$ in our modified basis fields implies an existing $e^{\pm i(gz-\Omega t)}$ offset, which must be accounted for in our ansatz in order to account for all even Fourier components. It is worth remarking that the possibility of block-diagonalizing the problem in this fashion is unusual for a photonic crystal. It is

owed to the fact that, as opposed to the discrete symmetries present in a conventional crystal, screw symmetry is a continuous symmetry. Hence, the change induced on the fields by an infinitesimal perturbation which respects this symmetry can always be recovered by applying the same symmetry operation to the fields themselves, which is the essence of Eqs. (8)–(11).

**Analytic band structures.** In the impedance-matched ($\alpha_\varepsilon = \alpha_\mu$ and $\phi = 0$) case, the 4-by-4 system above further decouples forward and backward-propagating waves, so that the off-diagonal matrices $\mathbb{M}_{\rightarrow,n}^{\leftarrow}$ and $\mathbb{M}_{\leftarrow,n}^{\rightarrow}$ vanish, as expected due to the impedance-matching condition. In this case, the eigenvalues can be easily calculated by hand. The eigenvalues for the $\phi = 0$ case can thus be written as:

$$\omega_{n,\pm}^{\rightleftarrows}(k) = -n\Omega + \sigma^{\rightleftarrows}\bar{k}_n\left(1 + \frac{\bar{\alpha}^+}{2}\right) \pm \sqrt{\left(\bar{k}_n\frac{\bar{\alpha}^+}{2}\right)^2 + \Delta_0^{\rightleftarrows}} \tag{13}$$

where $\bar{\alpha}^\pm = \bar{\alpha}_\varepsilon \pm \bar{\alpha}_\mu$ are the sum ( + ) and difference ( − ) between the electric and magnetic modulations, $\bar{\alpha}_{\varepsilon/\mu} = -\alpha_{\varepsilon/\mu}/(1 + 2\alpha_{\varepsilon/\mu})$, $k_n = k + ng$, $\Delta_0^{\rightleftarrows} = [(\bar{g} - \sigma^{\rightleftarrows}\Omega) + \bar{\alpha}^+\bar{g}](\bar{g} - \sigma^{\rightleftarrows}\Omega)$, while $\sigma^{\rightarrow} = +1$ and $\sigma^{\leftarrow} = -1$ stand for forward and backward-travelling modes. Note that in the impedance-matched case we have $\alpha^- = \bar{\alpha}^- = 0$. In Fig. 2 we show the analytic (lines) and numerical (triangles/circles for RHP/LHP) forward-propagating bands of first and second order, in order to show their interaction, as well as the first-order backward-propagating band (which only interacts weakly with the screw in this case) for increasing values of $\Omega$. Details of the numerical Floquet-Bloch calculations are given in the SM. Throughout the paper, we use $g = \varepsilon_1 = \mu_1 = 1$, so that the temporal modulation frequency $\Omega$ corresponds numerically to the screw velocity $v_s$, and we refer to them equivalently. Here we use $\alpha_\varepsilon = \alpha_\mu = \alpha = 0.4$. Note how the two bands display opposite circular polarisation in their fundamental harmonic, which we define as right-hand-polarized (RHP, blue) or left-hand polarized (LHP, red) according to the fixed-position/varying-time convention.

At low velocities (a-c), no band gaps are present in this impedance-matched scenario, as expected. Note how the first and second forward bands with the same polarization approach one another as $\Omega$ is increased. In fact, as $\Omega/g$ approaches a critical value $\Omega_{crit}^- = 1/(1 + 2\alpha)$ a transition occurs (in this instance $\Omega_{crit}^- = 0.5556$), with the appearance of a diagonal band-gap, hosting growing and decaying states, bounded by two exceptional points (d). This unstable gap closes again at the upper critical value $\Omega_{crit}^+ = 1$. These boundaries can be easily shown by studying the argument of the square root in the eigenvalues above. The complex pairs of states in the unstable phase are marked with a dashed line, located at $\Re[\omega_n(k)] + \Im[\omega_n(k)]$. The appearance of the unstable band-gap within the impedance-matched regime is a peculiar feature of luminal systems, which has been pointed out before[29]. However, the amplification mechanism in previous works did not manifest itself as a pair of exceptional points separated by a PT-broken phase with complex eigenvalues as in this case, but rather on the generation of a supercontinuum. Therefore, this is the first instance of such PT-symmetry breaking occurring near the luminal regime in spite of impedance-matching, which would normally be expected to prevent the formation of band-gaps. Note, furthermore, how this instability only occurs for one circular polarization, whereas states with the opposite polarization retain real eigenvalues.

In Fig. 3 we investigate the changes to the bands at long-wavelengths as we vary $\Omega$ between the two critical values $\Omega_{crit}^-$ and $\Omega_{crit}^+$, which bound the velocity regime within which complex states can be found. One characteristic feature of space-time media which was recently discovered is their ability to exert a

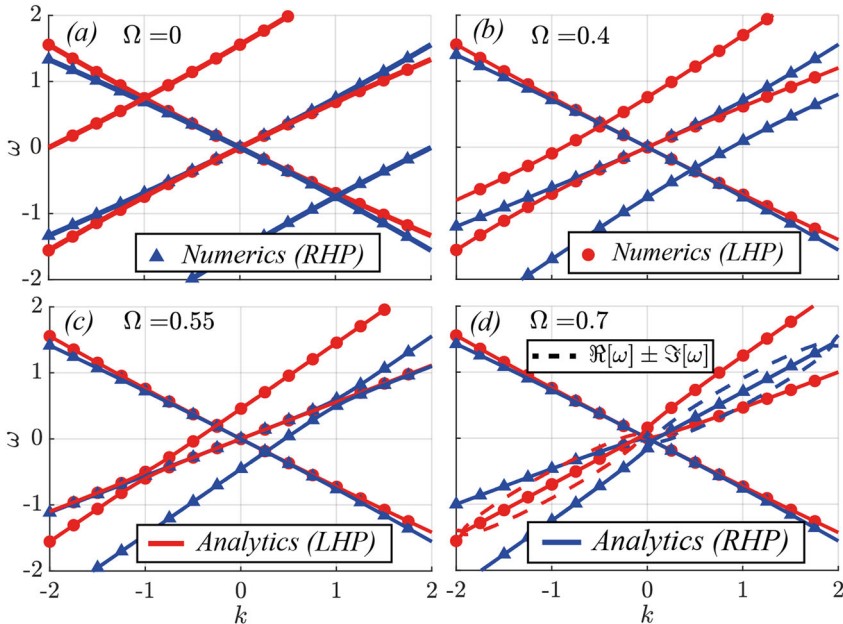

**Fig. 2 Band structure of the photonic Archimedes' Screw for the zero-dephasing ($\phi = 0$) and $\alpha_\varepsilon = \alpha_\mu = \alpha = 0.4$ (impedance-matched) case, for $g = 1$ (so that $v_s = \Omega$).** The different panels correspond to increasing modulation frequency/speed (**a**) $\Omega = 0$, (**b**), $\Omega = 0.4$, (**c**) $\Omega = 0.55$ and (**d**) $\Omega = 0.7$. Note the attraction (**c**) between forward RHP bands: as $\Omega \to \Omega_{crit}^- \approx 0.5556$, this interaction gives rise to the PT-broken phase, with complex RHP bands (the dashed lines show the two complex states $\Re[\omega] \pm \Im[\omega]$) being shown for $\Omega = 0.7$ in (**d**).

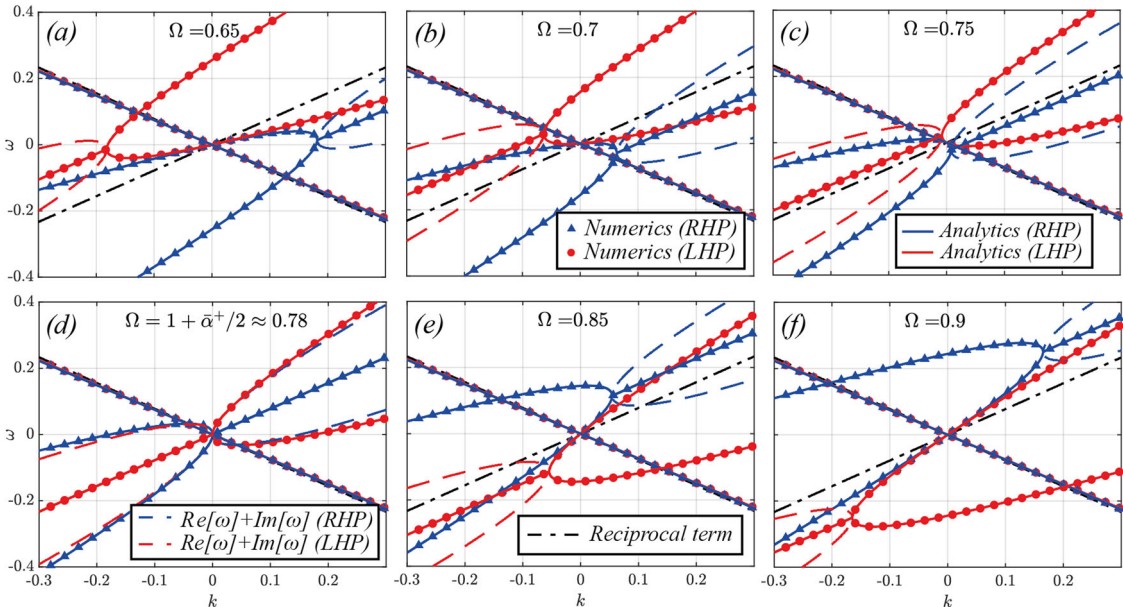

**Fig. 3 Band structure of the photonic Archimedes' Screw near the origin for the zero-dephasing (impedance-matched) case across the luminal $v_s \to 1$ regime.** Continuous lines denote analytic solutions for the lowest LHP (red) and RHP (blue) bands, and dots (LHP) and triangles (RHP) of the respective colours correspond to numerical simulations. The different panels correspond to different modulation frequencies (and velocities) across the luminal regime (**a**) $\Omega = 0.65$, (**b**) $\Omega = 0.7$, (**c**) $\Omega = 0.75$, (**d**) $\Omega = \Omega_{crit}^0 = 1 + \bar{\alpha}^+/2 \approx 0.78$, (**e**) $\Omega = 0.85$ and (**f**) $\Omega = 0.9$. Note how the exceptional points for the RHP bands in the first quadrant reaches the origin for $\Omega = \Omega_{crit}^0$, giving rise to a broadband chiral PT-broken phase. In addition, at this critical point the optical drag flips sign from negative (opposing the flow of light) to positive.

drag on light, in analogy with the Fresnel drag exerted by a moving medium, but with a wider tunability and with the possibility of superluminal modulation. This drag manifests itself as an asymmetry between the velocity of the forward and backward waves in the long-wavelength/low frequency limit

$k \ll g \wedge \omega \ll \Omega$. By Taylor-expanding the $\omega(k)$ eigenvalues above, we can clearly see that the first-order contribution: $\omega^{\rightleftharpoons}(k) = \pm(1 + \bar{\alpha}^+/2)k + O(k^2)$ yields the same speed for both propagation directions, and for both polarizations. This first-order contribution is depicted as dot-dashed black lines in Fig. 3,

which helps us visualise the optical drag induced by the screw as a result of the higher-order contributions. It is easy to see how forward waves of both polarizations are being dragged backwards as $\Omega$ increases towards another key critical value $\Omega_{crit}^0 = 1 + \bar{\alpha}^+/2$ ($\approx 0.78$ in panels a-c).

As $\Omega \to \Omega_{crit}^0$, something quite spectacular happens: the velocity of both RHP and LHP waves first tends to zero near the origin, the waves being effectively led to a halt by the screw, then becomes negative and tends to $-\infty$, implying that the optical drag becomes now infinite and opposite to the direction of the screw. In addition, while LHP states remain stable, the instability affecting RHP modes reaches the origin, implying a bandwidth-unlimited instability, so that the system can now amplify RHP waves of any frequency. It is important to stress that, in sharp contrast to previously studied luminal instabilities[28,29], this amplification mechanism preserves and amplifies the original frequency of a wave without generating a supercontinuum, as we show later in the Paper. Finally, for $\Omega_{crit}^0 < \Omega < 1$, the drag suddenly flips sign, tending to $+\infty$ as $\Omega \to \Omega_{crit}^0$ from above. This occurs after the exceptional point touches the origin, so that the opposite PT-exact branch is now pinned to it. Thus, forward waves now travel faster than backward ones, and the Screw is exerting a positive drag on the waves. Note how the velocities of the two polarizations are pinned together at the origin. This is a consequence of the PT-symmetry underlying this system: since a PT operation must flip the two polarizations into each other along with the $k$ and $\omega$ axes, the requirement that the bands be analytic near the origin implies that their slopes must be identical in its proximity.

With some more algebra, we can also derive the closed-form dispersion relation for the $\phi = \pi/4$ case, which reads:

$$\omega_{n,\pm}^{\rightleftharpoons} = -n\Omega \mp \bar{g}\left(1 + \frac{\bar{\alpha}_+}{2}\right)$$
$$+ \sigma^{\rightleftharpoons}\sqrt{\left(1 + \frac{\bar{\alpha}^+}{2}\right)\bar{k}_n^2 \pm 2(1 + \bar{\alpha}^+)\Omega\bar{k}_n + \Delta_{\pi/4}} \quad (14)$$

where $\Delta_{\pi/4} = \left(\frac{\bar{\alpha}^+\bar{g}}{2}\right)^2 + \Omega^2$ and $n$ is an odd integer. Note that in this case the offset between the modulations in $\hat{\varepsilon}$ and $\hat{\mu}$ implies that the system is no longer impedance-matched. As a result, band gaps are present at any screw velocity. In Fig. 4 we investigate the bands near the origin as we sweep over different modulation frequencies (velocities) $\Omega$ for $\alpha = 0.4$.

Despite having changed only the dephasing parameter $\phi$, we notice that the bands are remarkably different from the matched case, highlighting the richness and diversity of physics at play in this system. Firstly, note how, as opposed to the $\phi = 0$ case, here it is the forward waves which are hardly interacting with the screw, whereas the backward bands are dramatically altered. Secondly, note how the band-gaps here are conventional $\omega$-gaps and $k$-gaps, and no diagonal gaps are observed. The occurrence of $k$-gaps is well-known in the literature on time-varying media, and it is a signature of parametric amplification. However, $k$-gaps have only been observed in superluminal regimes, whereby the modulation travels faster than the waves in the pristine medium. On the contrary, this system is the first studied one (to the best knowledge of the authors) to exhibit $k$-gaps at modulation speeds well below $c$. Our analytic solution allows us to calculate the exact screw velocity $\Omega/g = \sqrt{1 + \bar{\alpha}^+}$ at which the $k$-gaps open (exceptional point), which in this case is 0.7454. Similarly to the $\phi = 0$ case, the second critical point is at $\Omega_{crit}^0 = 1 + \bar{\alpha}^+/2$, and it coincides with the positive and negative complex bands touching each other at the origin (panel d). Note that, concurrently with this critical point, the unstable band occurs precisely at $\omega = 0$ since the first two terms in the lowest eigenvalues cancel out, and the square root returns an imaginary number. Thus, this regime hosts a DC instability. Once again, this instability is chiral in nature, as only one of the two polarizations is affected by it, whereas the other is only subject to an optical drag.

In order to study in more depth the asymptotic slope of the bands at long wavelengths, in Fig. 5 we plot the velocity of the

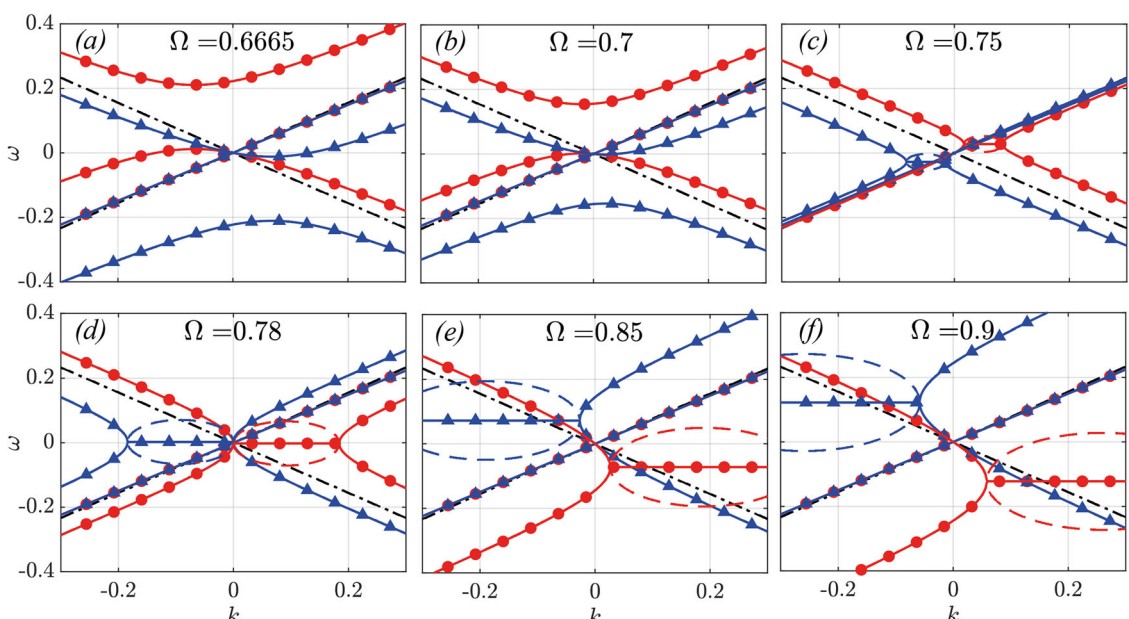

**Fig. 4 Band structures across the luminal regime for the $\phi = \pi/4$ case.** As in the previous scenario $\alpha = 0.4$ and the colour/marker scheme is identical. The different panels correspond to modulation frequencies (and velocities) (**a**) $\Omega = 0.6665$, (**b**) $\Omega = 0.7$, (**c**) $\Omega = 0.75$, (**d**) $\Omega = 0.78 \approx \Omega_{crit}^0$, (**e**) $\Omega = 0.85$ and (**f**) $\Omega = 0.9$. Note the qualitative difference with the $\phi = 0$ case in the character of the complex bands. Note how the band gaps here are either purely vertical or purely horizontal. Moreover, $k$-gaps in this system appear below the luminal limit, in contrast to all previous observations, where they only appear in superluminal scenarios.

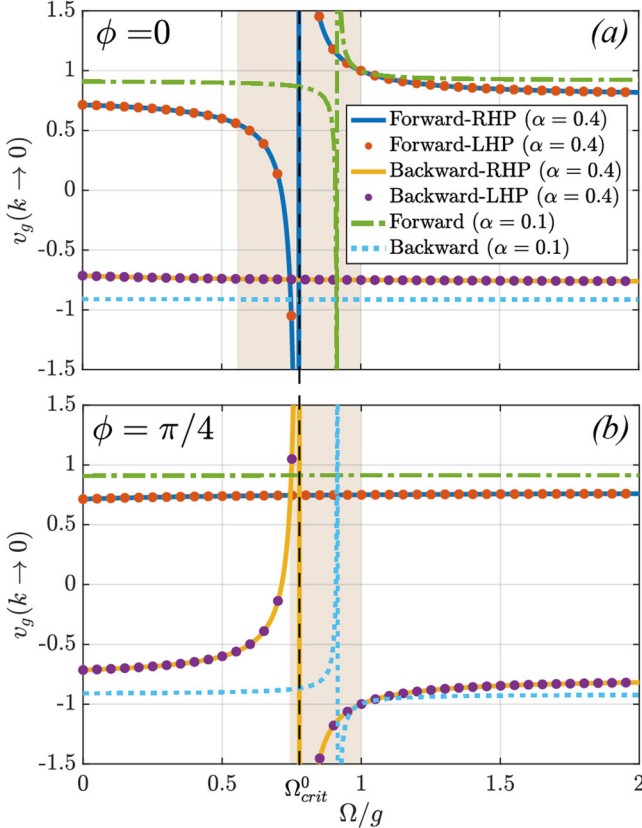

**Fig. 5 Long-wavelength (k → 0) velocity of the bands across all velocity regimes for half-dephasing (a) φ = 0 and (b) φ = π/4.** Continuous lines and circles are for $\alpha = 0.4$, whereas the dashed and dotted lines are for $\alpha = 0.1$, demonstrating the shrinking of the velocity range over which the interaction with the screw is strongest. Note that for $\phi = 0$ the screw interacts significantly only with the forward bands, whereas the opposite occurs for $\phi = \pi/4$. Note that the slopes of RHP and LHP bands are always degenerate in the $k \to 0$ limit, as a consequence of PT symmetry. Shaded regions correspond to velocity regimes comprised between the exceptional points $\Omega_{crit}^-$ and $\Omega_{crit}^+$ (for the $\alpha = 0.4$ case), while the critical point $\Omega_{crit}^0$ (also for $\alpha = 0.4$), common to both the $\phi = 0$ and $\phi = \pi/2$ cases, is marked as a black, dashed line.

forward and backward waves in the limit $k \to 0$ as a function of the screw velocity, for the $\phi = 0$ (top panel) and $\phi = \pi/4$ (bottom panel) cases, and for $\alpha = 0.4$ (continuous lines and dots) and $\alpha = 0.1$ (dashed and dot-dashed lines). For $\phi = 0$ it is evident that the forward waves are the only ones to be significantly affected by the screw. As $\Omega$ approaches the critical value $\Omega_{crit}^0$, the velocity of the forward waves decreases to the point where it becomes negative and tends to $-\infty$ with a resonance-like profile, as expected from our discussion of Fig. 3. This divergence and sign-flipping of the long-wavelength velocity occurs at $\Omega = 1 + \bar{\alpha}^+/2$, and is common to both the $\phi = 0$ and the $\phi = \pi/4$ cases. For the $\phi = \pi/4$ case, the situation is inverted, with the backward waves slowing down with $\Omega$ and flipping to forward-travelling ones, diverging at the above critical velocity and re-emerging as fast backward waves again after the transition. From these plots it is even more evident to observe how the two polarizations (lines for RHP, dots for LHP, shown for $\alpha = 0.4$ only) share the same velocity at the origin, as per our PT-symmetry argument above.

**Chiral instabilities.** We now turn our attention to the chiral instabilities observed in the band diagram for $\phi = 0$. In Fig. 6 we show the opening of the diagonal band gap as we vary $\Omega$ for fixed $\alpha = 0.4$ (left column) and as we vary $\alpha$ for fixed $\Omega = 0.6$. It can clearly be seen how, for RHP waves, the first two bands attract one another, to give rise to two complex solutions within the interval, bound by a pair of exceptional points, a well-known signature of PT-symmetry breaking[55]. In fact, while the system under study is only characterized by real material parameters, its time-dependence allows for the existence of PT-broken phases, where the material can amplify incoming waves. Crucially, the chiral nature of the optical Archimedes' screw implies that this system selectively amplifies RHP waves, as we set out to demonstrate.

In Fig. 7 we plot the transmission of incoming circularly polarized plane waves through a finite length $d$ of optical Archimedes' screw by plotting the Lissajous figure described by a cycle of the outgoing wave as a result of the beating between the frequency $\omega = 1$ of the incoming waves and the rotation frequency $\Omega$ of the screw. We plot the real part of the $x$ and $y$-components of the electric field. The left and right columns correspond to LHP and RHP input waves respectively. For LHP input, the top panel considers $\Omega = 1$, while the bottom panel assumes $\Omega = 0.5$, and the blue, red, yellow and purple curves correspond to different thicknesses $d = 0, 0.2\pi, 0.35\pi$ and $0.5\pi$, chosen to illustrate the change in the resulting Lissajous figure. Note that, as expected, the time required for the waves to complete a cycle of Lissajous figure corresponds to the beating time between the frequency $\omega = 1$ of the incoming waves and that of the screw $\Omega$.

For LHP waves, the interaction between the incoming wave and two real eigenstates simply results in Lissajous figures for the outgoing waves, whose complexity increases with the least common multiple between the two frequencies $\omega$ and $\Omega$, and no net gain is achieved. This picture changes dramatically for RHP waves: now the two states which the incoming wave couples to share the same real part, which results in simpler beating patterns. However, the imaginary part of the eigenvalues soon causes a net amplification of the outgoing waves as $d$ increases, a signature of the chiral nature of this amplification process. Note how the curves corresponding to increasing values of $d = 0, \pi/2, \pi$ and $3\pi/2$ result in the electric field to acquire larger and larger values, the curves moving farther away from the origin. The top right panel corresponds to $\Omega = 1$ and the bottom one to $\Omega = 0.75$.

Finally, in order to further investigate the underlying amplification mechanism at work, in Fig. 8 we plot the modulus of the $x$-component of the electric field as a function of time, over a period $2\pi/\Omega$ of the screw. We consider incoming waves of frequency $\omega = 1$, temporal screw frequency $\Omega = 0.8$ and dephasing $\phi = 0$. Panel (a) shows the dynamics of the transmitted field in the stable phase for small $\alpha = 0.05$. The interaction with the screw leads to a beating between the two, with a periodic exchange of energy between the screw and the waves, whose extent oscillates periodically with the thickness $d$ of the screw. By contrast, panel (b) shows the results for the unstable case $\alpha = 0.4$, where the incoming waves are exciting the unstable states in the PT-broken phase.

Orientation of the light polarisation is crucial to its coupling to a moving grating and to whether it extracts energy from or delivers energy to the grating. At low and high grating speeds the polarisation wave travels at a velocity markedly different from that of the grating and drifts through alternately amplifying and attenuating regions. Energy exchange oscillates up and down but

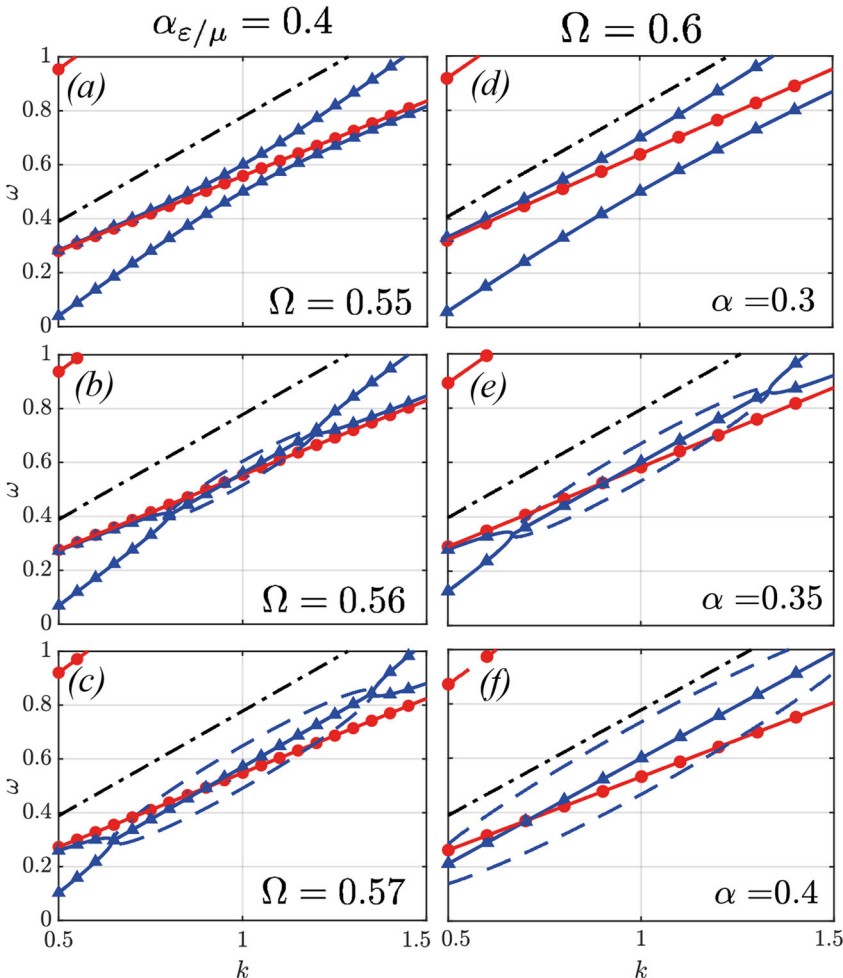

**Fig. 6 Band-gap opening for** $\phi = 0$**.** In the left column (**a-c**) we vary $\Omega$ from 0.55 (**a**) to 0.57 (**c**) at a fixed $\alpha = 0.4$ and in the right column we vary $\alpha$ from 0.3 (**d**) to 0.4 (**f**) ad a fixed $\Omega = 0.6$. Notice how both parameters induce the attraction between the two bands that gives rise to the transition into the PT-broken phase.

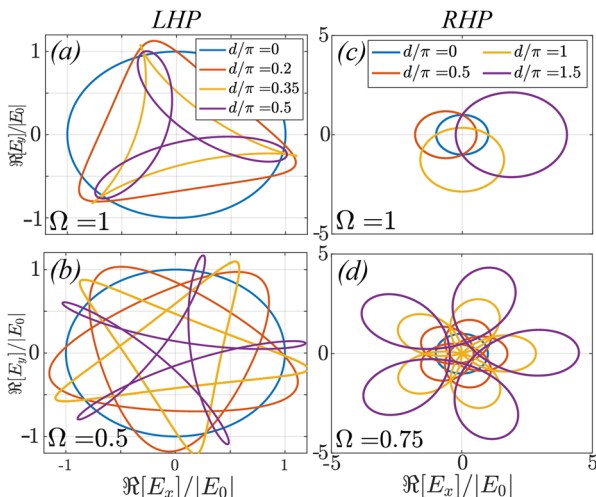

**Fig. 7 The x and y components of the transmitted electric field form a Lissajous figure, which originates from the beating between the frequency of the input wave** $\omega = 1$ **and that of the screw** $\Omega$**.** The commensurability ratio between the two determines the number of lobes in the figure. For RHP input (**a-b**) the screw is able to amplify the waves as they propagate through a thickness $d$. By contrast, a LHP input wave (**c-d**) is not amplified, but additional beating results from the additional, distinct real eigenvalue.

averages to zero and PT symmetry rules. On the other hand when the two speeds are comparable the polarisation has a means of locking its velocity to that of the grating: it can choose an orientation such that the local velocity, as determined by its overlap with the grating, is equal to that of the grating, thus maintaining its relative orientation as they move together, as evidenced by Fig. 8b. There are two orientations where this might happen; one is in the gain region, the other in the loss region. This gives rise to the band gap seen in figure 6 where we have two solutions, one gaining energy in time, the other losing energy. This mechanism is possible only over a range of velocities dictated by the amplitude of the grating. We contend that this grabbing hold of the light to raise its energy level is analogous to the function of an Archimedes screw in raising the level of water. It is worth remarking that, although the problem is mathematically more amenable in the impedance-matched case, this amplification effect is not contingent on both $\varepsilon$ and $\mu$ being modulated. In Supplementary Figure 1 we demonstrate this, showing the case where only $\varepsilon$ is modulated, as most easily accomplished in pump-probe experiments.

In this work we introduced chiral space-time metamaterials as an electromagnetic analogue of the Archimedes' Screw for fluids, investigating the exotic properties of their photonic bands by developing an analytic model which uncovered exact closed-form solutions to Maxwell's Equations, which we benchmarked against numerical calculations showing perfect agreement. Furthermore,

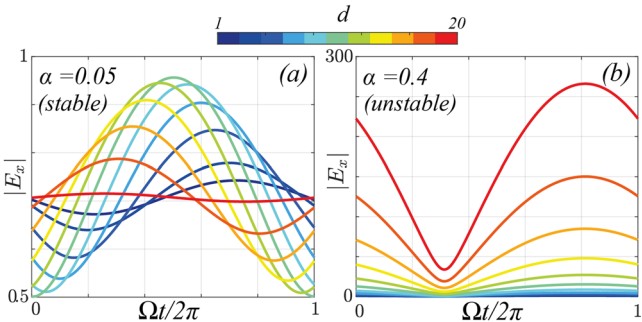

**Fig. 8 Modulus of the transmitted electric field as a function of time in (a) the stable phase ($\alpha = 0.05$) and (b) the unstable phase ($\alpha = 0.4$).** In the stable phase, the screw is not capable of capturing the wave's polarisation, which results in periodic oscillations. In the unstable phase, the polarisation of the wave is interlocked with that of the screw, which can thus grab hold of the polarisation of the waves and amplify them as they propagate through it. Here we use $\omega = 1$, $\Omega = 0.8$, $g = 1$ and $\phi = 0$.

we investigated the instabilities arising in these systems, and demonstrated their ability to amplify light of a specific polarization. The richness of the model presented offers plenty of opportunities for further investigation of the broad parameter space available, and the combination of time-dependence and chirality makes this new direction a promising ground for future studies of topological and non-Hermitian physics, and we envision that much of the physics at play can already be tested in optics with pump-probe experiments in highly nonlinear epsilon-near-zero materials, and at RF with nonlinear inductors and capacitors.

## Methods
Full details on all analytical and numerical methods used are freely available in the Supplementary Information, and could not be included in the main manuscript due to formatting limitations.

## Data availability
The main data supporting the findings of this study are available within the article and its Supplementary Information files. All the raw data generated in this study are available from the corresponding authors upon reasonable request. Requests will be dealt with by E.G. within a maximum timeframe of two weeks. Data will be provided under guarantee of acknowledgement/appropriate citation of this work and a scientifically sound reason for request.

## Code availability
All the data analysis codes related to this study are available from the corresponding authors upon reasonable request. Requests will be dealt with by E.G. within a maximum timeframe of two weeks. Codes used to produce the data will be provided under guarantee of acknowledgements/appropriate citation of this work, and a scientifically sound reason for request.

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

# ARTICLE

35. Lira, H., Yu, Z., Fan, S. & Lipson, M. Electrically driven nonreciprocity induced by interband photonic transition on a silicon chip. *Phys. Rev. Lett.* **109**, 033901 (2012).
36. Camacho, M., Edwards, B. & Engheta, N. Achieving asymmetry and trapping in diffusion with spatiotemporal metamaterials. *Nat. Commun.* **11**, 1 (2020).
37. Shaltout A. M., Shalaev & M. L. Brongersma, Spatiotemporal light control with active metasurfaces. *Science* **364**, eaat3100 (2019).
38. Alam, M. Z., De Leon, I. & Boyd, R. W. Large optical nonlinearity of indium tin oxide in its epsilon-near-zero region. *Science* **352**, 795 (2016).
39. Vezzoli, S. Optical time reversal from time-dependent epsilon-near-zero media. *Phys. Rev. Lett.* **120**, 043902 (2018).
40. Pacheco-Peña, V. & Engheta, N. Effective medium concept in temporal metamaterials. *Nanophotonics* **9**, 379 (2020).
41. Torrent, D. Strong spatial dispersion in time-modulated dielectric media. *Phys. Rev. B* **102**, 214202 (2020).
42. Huidobro, P., Silveirinha, M., Galiffi, E. & Pendry, J. Homogenization theory of space-time metamaterials. *Phys. Rev. Appl.* **16**, 014044 (2021).
43. Jaggard, D., Mickelson, A. & Papas, C. On electromagnetic waves in chiral media. *Appl. Phys.* **18**, 211 (1979).
44. Wang, B., Zhou, J., Koschny, T., Kafesaki, M. & Soukoulis, C. M. Chiral metamaterials: simulations and experiments. *J. Opt. A: Pure Appl. Opt.* **11**, 114003 (2009).
45. Lakhtakia, A. & Weiglhofer, W. S. On light propagation in helicoidal bianisotropic mediums. *Proc. R. Soc. Lond. Ser. A: Math. Phys. Sci.* **448**, 419 (1995).
46. Lindell, I., Sihvola, A., Tretyakov, S. & Viitanen, A. J. Electromagnetic waves in chiral and bi-isotropic media (Artech House, 1994).
47. Pendry, J. A chiral route to negative refraction. *Science* **306**, 1353 (2004).
48. Tretyakov, S., Nefedov, I., Sihvola, A., Maslovski, S. & Simovski, C. Waves and energy in chiral nihility. *J. Electromagn. waves Appl.* **17**, 695 (2003).
49. Zhang, S., Park, Y.-S., Li, J., Lu, X., Zhang, W. & Zhang, X. Negative refractive index in chiral metamaterials. *Phys. Rev. Lett.* **102**, 023901 (2009).
50. Rogacheva, A., Fedotov, V., Schwanecke, A. & Zheludev, N. Giant gyrotropy due to electromagnetic-field coupling in a bilayered chiral structure. *Phys. Rev. Lett.* **97**, 177401 (2006).
51. Menzel, C. Asymmetric transmission of linearly polarized light at optical metamaterials. *Phys. Rev. Lett.* **104**, 253902 (2010).
52. Wang, Z., Cheng, F., Winsor, T. & Liu, Y. Optical chiral metamaterials: a review of the fundamentals, fabrication methods and applications. *Nanotechnology* **27**, 412001 (2016).
53. Fernandez-Corbaton, I. New twists of 3d chiral metamaterials. *Adv. Mater.* **31**, 1807742 (2019).
54. Xiao, M., Lin, Q. & Fan, S. Hyperbolic weyl point in reciprocal chiral metamaterials. *Phys. Rev. Lett.* **117**, 057401 (2016).
55. Bender, C. M. *PT symmetry: In quantum and classical physics* (World Scientific, 2019).
56. Adobe Stock, 86671765 (2021).

## Acknowledgements

E.G. acknowledges funding from the EPSRC via the Centre for Doctoral Training in Theory and Simulation of Materials (Grant No. EP/L015579/1), an EPSRC Doctoral Prize Fellowship (Grant No. EP/T51780X/1) and a Junior Fellowship of the Simons Society of Fellows (855344,EG). P.A.H. acknowledges funding from Fundação para a Ciência e a Tecnologia and Instituto de Telecomunicações under projects HelicalMETA, UIDB/50008/2020 and the CEEC Individual program with reference CEECIND/02947/2020. J.B.P. acknowledges funding from the Gordon and Betty Moore Foundation.

## Author contributions

J.B.P. designed the project, E.G. carried out analytical and numerical calculations, P.A.H. and J.B.P. supervised the project, E.G. drafted the initial manuscript and all authors contributed to the final version of the paper.

## Competing interests

The authors declare no competing interests.
