## [Peer Review File · Nature Communications]

An Archimedes' Screw for LightREVIEWER COMMENTS

Reviewer #1 (Remarks to the Author):

The authors have introduced a new class of spatio-temporal media as chiral spatio-temporal media by introducing a specific form of spatio-temporally rotating media. Analytic closed form solutions have been obtained for its dispersion, showing its exotic property in amplifying light into a specific polarization. The work is interesting that PT-symmetry can be used to explain the instability and surprisingly closed form solution can be obtained. I believe the current work sets an excellent example on how spatio-temporally varying media can give rise to very special wave phenomena down to the fundamental level. The incorporation of polarization into time-varying media also points to new applications. I can recommend publication given the following (minor) comments are addressed.

1. In Fig. 5, can you also mark where the EP (or critical point) lies?
2. In Fig. 7, I appreciate the discussion of a system of finite length instead of infinite length, pointing to possible experiment. In this case, will it be appropriate to talk about critical Ω ? What will be the diagram look like?
3. Please discuss on possible experimental implementation. The medium seems require both rotating ϵ s and μ .
4. On the comparison to Archimedes' screw: is it just based on the resemblance of the spiral rotating action schematically drawn in Fig. 1 or it has a deeper connection in terms of functionality?

Reviewer #2 (Remarks to the Author):

This is an interesting article, investigating a new type of time varying electromagnetic material. The permittivity and permeability are anisotropic, with a principal axis that traces out a helix around the z axis. This helix moves in time with a velocity that can be chosen, and the phase of the helical precession can differ between ϵ and μ . The authors find that such media can exhibit a band structure that is very different for the two handedness of polarization, as well as providing polarization selective amplification.

Overall the paper is easy to read, and the theory is clear. I think the following points should be addressed to improve the manuscript:

(1) The term 'chirality' is used here. While the system clearly is chiral, I think - to avoid confusion - the article should distinguish the inhomogeneous magnetodielectric media examined here from media with chiral constitutive relations, i.e. $D = \epsilon E + i \kappa H/c$.

(2) The connection to the Archimedes screw is intuitive and interesting, but can it be made a bit more precise? I think this would improve the understanding of the paper.

The mechanical screw transports a fluid against gravity through trapping it within a helix. It seems (from the description in e.g. Fig. 8) that there is a *similar* phenomenon here, which allows for the wave to be amplified. However, I found it difficult to link the brief description in the caption with the theory, and found the idea of 'grabbing' the polarization difficult to justify beyond it being an appealing picture. I suggest moving part of the caption of Fig. 8 into the main text and justifying the physical picture with some fundamental principles, or the earlier dispersion diagrams.

(3) Fig. 8 - I think the x axis labels are incorrect. Shouldn't it be $\Omega t/2\pi$?

(4) I found the transformation (8-11) interesting, and the fact that the coupling between the basis functions can be eliminated is an interesting result. In this case we have a continuous rather than discrete symmetry where for a fixed time we can translate by dz , and rotate by $d\theta$ and the system is identical. Could we use this symmetry to explain the simplification (12)?

(5) I would find the first set of dispersion plots easier to understand if I had a comparison with very low velocities $\Omega \sim 0$.

We thank the editor and the reviewers for their time and careful consideration of our work towards publication. We have included point-by-point responses to the reviewers' comments below (in blue). The consequent changes to the manuscript have also been marked in blue in the updated version, while any specific deletions are marked in red.

With kind regards,

Emanuele Galiffi, Paloma A. Huidobro and J. B. Pendry

Reviewer #1 (Remarks to the Author):

The authors have introduced a new class of spatio-temporal media as chiral spatio-temporal media by introducing a specific form of spatio-temporally rotating media. Analytic closed form solutions have been obtained for its dispersion, showing its exotic property in amplifying light into a specific polarization. The work is interesting that PT-symmetry can be used to explain the instability and surprisingly closed form solution can be obtained. I believe the current work sets an excellent example on how spatio-temporally varying media can give rise to very special wave phenomena down to the fundamental level. The incorporation of polarization into time-varying media also points to new applications. I can recommend publication given the following (minor) comments are addressed.

We thank Rev. 1 for their positive feedback and for recommending publication of our manuscript.

1. In Fig. 5, can you also mark where the EP (or critical point) lies?

We thank Rev. 1 for pointing us towards clarifying this in Fig. 5. We have now marked in Fig. 5 a shaded area corresponding to the velocity range between the EPs Ω_{crit}^- and Ω_{crit}^+ for the $\alpha = 0.4$ case in both the $\phi=0$ (a) and $\phi=\pi/4$ (b) cases, as well as the critical point Ω_{crit}^0 (which is common to both dephasing parameters) corresponding to the band-flipping.

2. In Fig. 7, I appreciate the discussion of a system of finite length instead of infinite length, pointing to possible experiment. In this case, will it be appropriate to talk about critical Ω ? What will be the diagram look like?

This is a very good point to reflect upon and clarify. Strictly speaking, the critical values derived in the first part of the paper apply to the infinite system. However, it is worth pointing out that the instability is of a "local" nature. This idea is clearer if combined with the explanation in Fig. 8, and the answer to comment 4 below: the trapping of the wave within a period of the screw, which is responsible for this effect, effectively decouples different periods of it. This is to be compared to what happens when there is a transient gradient in refractive index caused by a pump beam, which effectively traps a probe wave in its wake. As a result, it is not necessary for the length of the screw to span several modulation periods, which is confirmed by the fact that in Fig. 7d we effectively only see an exponential amplification of the field. However, we observed small signs of "finite-size" effects for very small "d" in Fig. 8b (blue curves at the bottom), whereby the localization point of the waves would adjust slightly before becoming fixed at one point within a period, from which point it is simply amplified, and the locations of its maxima and minima of intensity stay put. We have broadened our discussion around Figs. 7 and 8 in order to clarify this point.

3. Please discuss on possible experimental implementation. The medium seems require both rotating ϵ s and μ .

We thank Rev. 1 for the opportunity to make this important remark. Although this problem is most amenable analytically in the impedance-matched case, the amplification phenomenon is present also in the case where only ϵ is modulated. We have added a remark on this point before the concluding paragraph, in relation to Fig. 8, and an equivalent figure to the SM.

4. On the comparison to Archimedes' screw: is it just based on the resemblance of the spiral rotating action schematically drawn in Fig. 1 or it has a deeper connection in terms of functionality?

We thank Reviewer 1 for this useful question, which gave us an opportunity to explain further the analogy with an Archimedes' screw. Whilst there is a clear analogy between the two as depicted in Fig. 1, a deeper equivalence exists indeed in the unstable regimes: in a mechanical Archimedes' screw, the fluid finds an equilibrium position (the bottom of the helix) which is energetically favourable. The fluid is effectively trapped at the bottom of the helix, and must follow its movement. Similarly, in the unstable regime of the electromagnetic screw, where the wave interacts most strongly with the modulation, the phase of the wave becomes locked with the movement of the screw via its polarization, which provides the energetic well responsible for the trapping. We have now expanded our discussion of this analogy, in particular in the later part of the manuscript

Reviewer #2 (Remarks to the Author):

This is an interesting article, investigating a new type of time varying electromagnetic material. The permittivity and permeability are anisotropic, with a principal axis that traces out a helix around the z axis. This helix moves in time with a velocity that can be chosen, and the phase of the helical precession can differ between epsilon and mu. The authors find that such media can exhibit a band structure that is very different for the two handedness of polarization, as well as providing polarization selective amplification. Overall the paper is easy to read, and the theory is clear.

We thank Rev. 2 for their positive feedback on our work.

I think the following points should be addressed to improve the manuscript:

(1) The term 'chirality' is used here. While the system clearly is chiral, I think - to avoid confusion - the article should distinguish the inhomogeneous magnetodielectric media examined here from media with chiral constitutive relations, i.e. $D = \epsilon E + i \kappa H/c$.

We thank Rev. 2 for pointing out this potential confusion. We have now included a footnote where we specify that the system here does not include any bianisotropy.

(2) The connection to the Archimedes screw is intuitive and interesting, but can it be made a bit more precise? I think this would improve the understanding of the paper. The mechanical screw transports a fluid against gravity through trapping it within a helix. It seems (from the description in e.g. Fig. 8) that there is a *similar* phenomenon here, which allows for the wave to be amplified. However, I found it difficult to link the brief description in the caption with the theory, and found the idea of 'grabbing' the polarization difficult to justify beyond it being an appealing picture. I suggest moving part of the caption of Fig. 8 into the main text and justifying the physical picture with some fundamental principles, or the earlier dispersion diagrams.

We thank Rev. 2 for the positive comment. Orientation of the light polarisation is crucial to its coupling to a moving grating and to whether it extracts energy from or delivers energy to the grating. At low and high grating speeds the polarisation wave travels at velocities markedly different from that of the grating, and drifts through alternately amplifying and attenuating regions. Energy exchange oscillates up and down but averages to zero, so that PT symmetry rules. On the other hand, when the two speeds are comparable, the polarization has a means of locking its velocity to that of the grating: it can choose an orientation such that the local velocity, as determined by its overlap with the grating, is equal to that of the grating thus maintaining its relative orientation as they move together as evidenced by Fig. 8b. There are two orientations where this might happen; one is in the gain region, the other in the loss region. This gives rise to the band gap seen in figure 6 where we have two

solutions, one gaining energy in time, the other losing energy. This mechanism is possible only over a range of velocities dictated by the amplitude of the grating. We contend that this grabbing hold of the light to raise its energy level is analogous to the function of an Archimedes screw in raising the level of water.

(3) Fig. 8 - I think the x axis labels are incorrect. Shouldn't it be $\Omega t/2\pi$?
Indeed, thank you for pointing this out. We have now fixed this typo.

(4) I found the transformation (8-11) interesting, and the fact that the coupling between the basis functions can be eliminated is an interesting result. In this case we have a continuous rather than discrete symmetry where for a fixed time we can translate by dz , and rotate by $d\theta$ and the system is identical. Could we use this symmetry to explain the simplification (12)?

We thank Rev. 2 for pointing out this neat way of explaining the fact that we can block-diagonalize the system. Indeed, as opposed to a typical photonic crystal, in this case there exists a continuous symmetry on top of the discrete ones, which is the essential ingredient which enables the decoupling of the Fourier modes. We have now included this as part of the discussion following Eq. 12.

(5) I would find the first set of dispersion plots easier to understand if I had a comparison with very low velocities $\Omega \sim 0$.

We thank Rev. 2 for pointing out this lack of clarity. We had originally started from $\Omega = 0.4$ in Fig. 2 since the larger changes in the band structure start occurring from those values onwards. Precisely for this reason, starting from $\Omega = 0$ as suggested by Rev. 2 is not a problem, and we agree that it puts the reader on firmer ground as a first figure from which to understand the rest. We have now substituted panel (a) to start varying Ω from 0. The difference is indeed minor.

REVIEWERS' COMMENTS

Reviewer #1 (Remarks to the Author):

The authors have addressed all of my concerns and comments. I am happy to support its publication.

Reviewer #2 (Remarks to the Author):

The authors have addressed all the points I raised, incorporating the suggested changes. I recommend the new version for publication.